# Optical Control of Superlattices States Formed Due to Electronic Phase Separation in Multiferroic Eu_0.8_Ce_0.2_Mn_2_O_5_

**DOI:** 10.3390/nano11071664

**Published:** 2021-06-24

**Authors:** Viktoriya Sanina, Boris Khannanov, Evgenii Golovenchits

**Affiliations:** Ioffe Institute, 194021 Saint Petersburg, Russia; khannanov@mail.ioffe.ru (B.K.); e.golovenchits@mail.ioffe.ru (E.G.)

**Keywords:** multiferroic, heterostructures–superlattices, optical pumping, cycling in a magnetic field

## Abstract

The effect of optical pumping and magnetic field on properties of the electronic phase separation regions, which are the multiferroic semiconductor heterostructures in the form of superlattices, have been studied in Eu_0.8_Ce_0.2_Mn_2_O_5_. These superlattices are formed due to self-organization in a dielectric crystal matrix as a result of the competing internal interactions balance and occupy a small crystal volume. The dynamical equilibrium states of superlattices are initially formed during cycling of as-grown samples in a magnetic field. The superlattices in such states are ferromagnetic and electrically neutral. Sets of ferromagnetic resonances were observed from individual layers of superlattices. Their features give rise information on properties of these layers and of a superlattice as a whole. The differences in the parameters of these resonances were due to different distributions of Mn^3+^ and Mn^4+^ ions in individual superlattices layers. It has been found that optical pumping having different powers allows us to control of multiferroic properties of superlattices layers by changing their magnetic and electric properties. It is shown that, under certain conditions, it is possible to significantly increase the temperatures at which multiferroic heterostructures exist.

## 1. Introduction

Multiferroics RMn_2_O_5_ (R are rare-earth ions, Bi) belong to type II multiferroics, in which ferroelectric ordering below the Curie temperature (T_C_ ≈ 30–35 K) is induced by magnetic ordering with the Néel temperature (T_N_ ≈ 35–40 K) [1,2]. At room temperature, they are described by the centrosymmetric sp.gr. Pbam. Due to close magnetic and ferroelectric ordering temperatures, such multiferroics exhibit the maximal magneto-electric effect. A characteristic feature of RMn_2_O_5_ is the presence of an equal number of Mn^3+^ ions (containing three t_2g_ and one e_g_ electrons on the 3d shell) and Mn^4+^ ions (with three t_2g_ electrons on the 3d shell), which provides conditions for charge ordering at a dielectric state. Mn^4+^ ions have an octahedral oxygen environment and are located in the layers with z = 0.25c and (1 − z) = 0.75c. Mn^3+^ ions have an off-center local environment in the form of pentagonal pyramids in the layers with z = 0.5c. R^3+^ ions with an environment similar to Mn^3+^ are in the layers with z = 0 [3]. The charge ordering and a finite probability of e_g_ electron transfer between Mn^3+^-Mn^4+^ ion pairs (double exchange) [4,5] are the key factors responsible for multiferroic properties at all temperatures. The low-temperature ferroelectric state at T ≤ T_C_ is mainly due to the charge ordering along the b-axis. The alternation of neighboring Mn^3+^ and Mn^4+^ ion pairs with strong ferromagnetic (double exchange) and weak indirect antiferromagnetic exchange along this axis gives rise to an exchange striction, which violates the lattice centrosymmetric state along the b axis and gives rise to the low-temperature ferroelectric ordering [6].

Note that studied by us the multiferroics-manganites are similar in their properties to the manganites LaAMnO3 (A = Sr, Ca, Ba) detail investigated before [7,8,9], that also contain Mn^3+^ and Mn^4+^ ions in various ratios depending on the degree of doping. Both types of manganites belong to systems, which are characterized by the strong electron correlations, exhibiting instability of uniform magnetic and charge orderings with respect to the formation of ferromagnetic and charge polarons (regions) with 100 Å–1000 Å sizes. In these regions, arranged in the dielectric antiferromagnetic matrix, a ferromagnetic orientation of spins Mn^3+^ and Mn^4+^ ions arises due to double exchange. Such local regions, containing electrons, recharging Mn^3+^ and Mn^4+^ ions, tend to be located far from each other, decreasing the energy of the Coulomb repulsion. As a result, these regions occupy a small volume in the crystal matrix [7,8,9].

In multiferroics-manganites, the finite probability of e_g_ electrons tunneling between Mn^3+^-Mn^4+^ ion pairs located in adjacent layers perpendicular to the c axis leads to formation of the similar electronic phase separation forming regions of nanoscopic sizes with the distribution of Mn^3+^ and Mn^4+^ ions differing from the initial crystal matrix. Such regions are simultaneously magnetic and ferroelectric ones possessing large magneto-electric interactions. They exist from the lowest temperatures to the temperatures well above the temperatures of multiferroic ordering.

We studied the phase separation states in RMn_2_O_5_ multiferroics and in doped R_0.8_Ce_0.2_Mn_2_O_5_ (R = Eu, Gd, Bi, Er, Tb) also [10,11,12,13,14,15,16,17,18,19,20,21,22,23,24]. Note that RMn_2_O_5_ and doped R_0.8_Ce_0.2_Mn_2_O_5_ crystals have the same Pbam central symmetry. Investigations of dielectric and magnetic properties, heat capacity, X-ray diffraction, Raman light scattering [10,11,12,13,14,15], electric polarization [15,16,17,18,19,20,21,22], and µSR studies [23,24] were carried out.

The regions in RMn_2_O_5_ and R_0.8_Ce_0.2_Mn_2_O_5_ (as in manganites LaAMnO_3_ (A = Sr, Ca, Ba)) are formed at the balance of the strong interactions (double exchange (with a characteristic energy of 0.3 eV), Jahn–Teller interaction (0.7 eV), and Coulomb repulsion (1 eV)) [5,7,8,9]. For this reason, they exist in a wide range of temperatures, from low ones to above room temperature [10,12,13,14,15,16,17,18,19,20,21,22], while having a large magneto-electric coupling.

The first two interactions contribute to the accumulation of electrons in the regions; the Coulomb repulsion ensures their equilibrium concentrations [5,7,13,25]. The layered structure of the Mn^3+^ and Mn^4+^ ion distribution perpendicular to the c axis in the main matrices of RMn_2_O_5_ and R_0.8_Ce_0.2_Mn_2_O_5_ crystals is also present in the distribution of Mn^3+^ and Mn^4+^ ions inside regions. As a result, a set of the superlattices arises in the crystal matrix. It can be presented in the form of periodically changing isotropic ferromagnetic layers with ferromagnetic boundaries between them which do not prevent of the e_g_ electron transport between the layers at double exchange. Figure 1 schematically shows a method for formation of ferromagnetic 1D superlattices (a) and an image of one of such superlattices (b) [25].

Sets of ferromagnetic resonances (FMR) were observed from individual layers of superlattices in EuMn_2_O_5_ (EMO) and Eu_0.8_Ce_0.2_Mn_2_O_5_ (ECMO). Their features give information on properties of these layers and of a superlattice as a whole [12,13,14]. It was found that the differences in the parameters of individual FMR lines of superlattices were due to different distributions of Mn^3+^ and Mn^4+^ ions in the individual superlattices layers. In EMO and ECMO, FMR lines of superlattices existed up to T < 60 K and T < 80 K, respectively. The sizes of these superlattices were 900 Å (EMO) and 700 Å (ECMO).

Thus, the study of a set of ferromagnetic resonances in the form of low-temperature responses of superlattices layers makes it possible to judge about the relatively low-temperature charge distribution of Mn^3+^ and Mn^4+^ ions in the superlattices layers. To confirm the existence of superlattices at room temperature, X-ray diffraction studies of the ECMO at this temperature have been presented [10].

X-ray diffraction studies were performed for several samples of ECMO at room temperature. The angular intensity distribution of 004 CuKα1 Bragg reflections was detected in the three-crystal regime with the ∆θ (ω/2ω − scan) having a resolution of 10″. As a monochromator and an analyzer, germanium crystals in the 004 reflection were used, which allowed conditions of nearly dispersion-free high-resolution survey geometry to be realized. Figure 1c shows diffraction curves for ECMO. As can be seen, Figure 1c exhibits a set of diffraction maxima separated into two pronounced regions. Each region has five to six peak pulsations, one of which is dominating in intensity. Both regions have similar widths and number of maxima. Analysis of the diffraction curve has led to the conclusion that the natures of the maxima are different. There are layer peaks whose positions are determined by the lattice parameter d (the upper axis of Figure 1c) and which characterize the layered crystal structure. The distances between the maxima of these peaks give information on the lattice mismatch in the layers. In addition, the diffraction curve exhibits periodic intensity oscillations with the d-independent period, pointing to the presence of a two-layer deformed area at the crystal surface. All five intensity maxima in the left-hand region of the diffraction curve are layer peaks, while only two maxima in the right-hand region are layer peaks, the remaining peaks being oscillatory. We attribute the more powerful right-hand peak to the original crystal matrix, which occupies the bulk of the crystal. The less intense left peak with a pronounced layered structure we attribute to superlattices formed by phase separation. This suggests that superlattices indeed exist in ECMO up to room temperature. As can be seen, in the left-hand region, which characterizes the superlattices, there are inhomogeneous responses intensities from different layers. The two outermost layers on the right side have increased intensity. The distribution of individual superlattices layers intensities makes it possible to judge the nature of their charge ordering at room temperature.

The balance of strong interactions indicated above ensures the formation of dynamic equilibrium states of superlattices. To control their properties, one (or several) interactions responsible for this balance should be subjected to changes. They are temperature variations, optical pumping of various powers, and application of magnetic field.

The main goal of the research presented in this paper was to study the behavior of superlattices states under such influences and to find methods for formation of the new states of superlattices existing to higher temperatures.

## 2. Objects and Research Methods

Single crystals of ECMO were grown by spontaneous crystallization from a solution-melt [26,27]. They were in the form of plates with a thickness of 1–3 mm and an area of 3–5 mm^2^. Natively faceted single crystals were used. The crystal symmetry and composition were determined by the X-ray phase analysis and X-ray fluorescence technique, respectively.

All crystals were in the form of plates with well-defined axes. The c axis was perpendicular to the plane of the plates, and the a axis orientation was clearly defined. The measurements were performed in the temperature range of 13–300 K at frequencies of 30–40 GHz in an applied magnetic field of up to 20 kOe.

Note that crystals containing rare earth ions Ce were characterized by a variable valence Ce^+3.75^, i.e., these ions exhibited the properties of both Ce^3+^ and Ce^4+^ ions. In [28], the existence of Ce ions with such a variable valence was revealed in studies of ceramic samples of Bi_0.9_Ce_0.1_Mn_2_O_5_. Ce ions in ECMO replace the Eu^3+^ ions. Recall that R ions in RMn_2_O_5_ have a pentagonal (distorted hexagonal) oxygen environment. According to [29], the ionic radii of the ions of interest (in a hexagonal environment) are as follows: Ce^4+^-0.87 Å, Ce^3+^- 1.01 Å, Eu^3+^-0.89 Å. Thus, it is more likely that Eu^3+^ ions can be replaced by Ce^4+^ ions and, with a lower probability, by Ce^3+^ ions.

The superlattices were also observed in the initial EMO crystals [10,13]. The doping with Ce^4+^ ions led to a significant increase in the concentration of neighboring pairs of Mn^3+^-Mn^4+^ ions in the planes perpendicular to the c axis. In ECMO, electrons in the z = 0 plane resulted from the reaction Eu^3+^ = Ce^4+^ + e. These electrons in the z = 0.25c and 1 − z = 0.75c planes converted Mn^4+^ ions into Mn^3+^ ions. As a result, the number of Mn^3+^-Mn^4+^ pairs and the concentration of superlattices were increased. However, as before, these superlattices occupied a small crystal volume in ECMO [10,13].

Some of the Eu^3+^ ions were replaced by larger Ce^3+^ ions (though with a lower probability). These ions contained lone pairs of 6s^2^ electrons on the outer shells. As it is known, the ions containing lone pairs of 6s^2^ electrons on the outer shells locally distort the lattice [30] thus forming a part of superlattices in ECMO with other properties.

To perform FMR measurements, we used a transmission type magnetic resonance spectrometer with a low-amplitude magnetic modulation. The measurements were carried out in the temperature range 13–300 K at frequencies 30–40 GHz in an applied magnetic field up to 20 kOe generated by an electromagnet. The cryostat with optical windows was located in the microwave channel ensuring a uniform distribution of the microwave field close to the sample. The microwave radiation (with the wave vector k) was directed along the c axis perpendicular to the platelet plane. The static magnetic field H was oriented along the a crystal axis and perpendicular to the direction of the microwave field h.

The FMR signals were amplified by Lock-in SR530 amplifier. Optical pumping was performed by a solid-state pulsed neodymium laser LTIPCH-8 with a simultaneous generation of the first (1.06 μm) and second (532 nm) harmonics. In this case, formation of excitons in the Mn ions subsystem occurred by the optimal way [31].

## 3. Experimental Data and Analysis

Before discussing the effect of optical pumping at various powers on properties of ECMO superlattices layers in the dynamically equilibrated states (which is the main goal of the research), let us consider the effect of temperature and magnetic field changes on the properties of such superlattices (studied in [10,11,12,13]) without of pumping. To form the equilibrium state of superlattices in EMO and ECMO, for whom the definite set of FMR lines was observed, it was necessary to produce the cycling of the as-grown samples in a magnetic field (successive field increase and decrease). The dynamically equilibrated state of the superlattices was established after a 3-fold cycling of the field in the 0–20 kOe interval. Note that if the sample, being in equilibrium state of superlattices, was heated to room temperature and kept at this temperature for a long time (several weeks), the equilibrium state of superlattices was reproduced at subsequent sample cooling to helium temperatures.

A set of FMR lines for dynamically equilibrated superlattices states in ECMO was observed along different crystal axes in the same magnetic fields. The studies of the FMR of superlattices layers for a number of RMn_2_O_5_ (R = Eu, Er, Tb, Gd) and R_0.8_Ce_0.2_Mn_2_O_5_ (R = Eu, Gd) crystals [12,13,14] showed that the magnitudes of magnetic fields, in which individual FMR lines were observed, differed insignificantly and did not depend on the R ions type. The Er^3+^ and Tb^3+^ ions (in contrast to Gd^3+^) strongly interacted with the lattice. This means that the superlattices states are determined by the internal interactions, the magnitude of which are significantly higher than the magnetic anisotropy fields, which had to be defined by the R ions type. Indeed, as noted above, the equilibrium states of the superlattices layers were determined by the balance of interactions in the subsystem of manganese ions (double exchange, Jahn–Teller, and Coulomb interactions). It was found that the differences in the parameters of individual FMR lines of superlattices layers were due to different distributions of Mn^3+^ and Mn^4+^ ions in individual layers of the superlattices, which were caused by the transfer of e_g_ electrons between these ion pairs. As a result, a set of superlattices layers could be presented in the form of periodically changing isotropic ferromagnetic layers with ferromagnetic boundaries between them, which did not forbid the transfer of e_g_ electrons between layers during double exchange (Figure 1b). Note that the responses from individual layers of the superlattices give rise the total response from ions of different valence located in each of the layers. In fact, we have a collective response from the layers of superlattices, the effective g-factor of which will differ from the g-factors of individual ions located in these layers.

It was also found that FMRs were recorded for superlattices in the frequency bands separated by the intervals in which FMRs were not observed. This was because the band states in the form of minibands separated by gaps were formed in superlattices (similar to the semiconductor superlattices [32]). The frequencies, at which the measurements were carried out (29–36 GHz), were corresponded one of the main minibands for ECMO. As noted above, temperature dependences of the FMR line intensities for dynamical equilibrium superlattices states in EMO and ECMO showed that superlattices in these crystals existed up to 60 and 85 K, respectively [13]. With increasing temperature, the growth in the kinetic energy of electron hopping conductivity change the equilibrium state of the superlattices. Electrons begin to populate layers with deeper wells at their boundaries (R1 and R2 layers) (Figure 1b). The number of superlattices layers decreases and their shape changes and transforms into the 2D structure [10,13]. Such two-dimensional structures exist up to much higher temperatures and give rise to a high-temperature electric polarization of these structures controlled by magnetic field [17,18,19,20,21,22].

Analysis of experimental data on properties of the superlattices in the dynamically equilibrated states in ECMO allowed us to build a model of spin wave excitations in the superlattices layers [13,14] based on the dispersion equations for spin waves of isotropic ferromagnetic films in 2D multilayers [33]. This model shows that the homogeneous spin-wave excitations with k = 0 (FMR) are excited in all layers of the superlattices and on their turning surfaces. The presence of turning surfaces near the superlattice layer edges makes ferromagnetic superlattices independent of the main antiferromagnetic matrix of multiferroics-manganites, and they are oriented by an external magnetic field. This is exactly what is observed in our case.

For the 1-mm-thick ECMO sample with an area of 5 mm^2^, the sequence of FMR lines in the equilibrium state is shown in Figure 2a. These lines correspond to those measured earlier in [12,13]. We can see five FMR lines schematically shown in Figure 1b and originating from different layers of superlattices. The most intense central L0 line appears in the applied magnetic field corresponding to the FMR at frequency 34.5 GHz for an isotropic ferromagnetic with the *g* factor *g* = 2. Less intense lines, to the left of the main resonance (L1 and L2) and to the right of (R1 and R2), are approximately symmetrically located in lower and higher magnetic fields with respect to the L0 line. In this case, the most intense lines (L1, L0, and R1) are doubled in contrast with measured in [12,13]. At somewhat lower fields, less intense wide lines are observed.

As one can see in Figure 3, frequencies of the FMR lines of superlattices layers linearly depend on the magnetic field and are described as
ω_LN_ = γ_LN_ (H_0_ + H_eff._^LN^)(1)

Here ω_LN_ are the circular frequencies for N lines (N = L1, L2, L0, R1, R2); γ_LN_ are the gyromagnetic ratios, which make it possible to determine the corresponding values of g- factors; H_eff._^LN^ are the internal effective fields leading to gaps on the ω_LN_ (H) dependences. As one can see, the H_eff._^LN^ values are positive for lines L1, L2 and increase with increasing number L. H_eff._^LN^ are negative for lines R1, R2. The g factors for the R1, R2 lines are slightly less than 2.0, while the g-factors of the L1, L2 lines are greater than 2.0. H_eff._^L0^ is equal to zero and the g-factor of the L0 line is exactly equal to 2.0.

This type of frequency dependence for the set of FMR lines can be understood if we assume that superlattices layers contain different numbers of Mn^3+^-Mn^4+^ ion pairs, and hence, different concentrations of e_g_ electrons that recharge these ion pairs. The L0 layer contains equal numbers of Mn^3+^ (donors) and Mn^4+^ (acceptors) ions, the charge ordering of which indicates that these layers are dielectric and are completely compensated semiconductor L0 layers. In this case, the Fermi level is located at the bandgap center [34]. Figure 4 shows the schematic energy diagram of such a layer.

Following the approach reported in [13,14], we analyze the properties of superlattices layers using the model of semiconductor heterostructures containing alternating L, 0, and R layers, in which Mn^4+^ and Mn^3+^ ions are treated as acceptor and donor impurities, respectively. Analyzing the characteristics of the observed FMR lines related to the layers of superlattices, we revealed the properties of these layers. As noted above, the L0 layers contain equal numbers of Mn^4+^ and Mn^3+^ ions, i.e., these layers are fully compensated semiconductors. In such layers, the Fermi level is located in the center of the band gap, and the insulating state arises in the L0 layers.

The FMR of L0 layers have the maximum intensity and minimum line width as compared with other layers (Figure 2a). The FMR of these layers is observed at frequency ω_0_ = γ_0_H, g = 2.0, H_eff_^L0^ ≈ 0 (Figure 3). The L2 and L1 lines states, in which the number of acceptor Mn^4+^ ions exceeds that of donor Mn^3+^ ions to different degrees (Figure 1b), are characterized by the holes type conductivity. In R2 and R1 layers, the number of Mn^3+^ ions exceeds that of the Mn^4+^ ions (Figure 1b) and electron conductivity prevails. Excessive Mn^3+^ ions are located in the octahedral positions of Mn^4+^ ions, i.e., they are Jahn–Teller ions providing the local lattice distortions in these layers. As a result, the distortions of deformed potentials inside these layers, the deepest wells of which cross the Fermi level and form electron drops, occur (Figure 4). The diamagnetic contribution of electron magnetization of these drops in an applied magnetic field H results in negative values of H_eff_^R1^, H_eff_^R2^ (Figure 3). To observe FMR from R1, R2 layers at a fixed frequency ω_0_, the external fields H must be increased by H_eff_^R1^, H_eff_^R2^ respectively (Figure 3). Excessive electrons also decrease the g-factors for R1, R2 lines to the values less than 2.0 and lead to the broadening of these lines. On the contrary, the Mn^4+^ ions and holes conductivity predominate in the L1 and L2 layers due to lower concentrations of Mn^3+^ ions and delocalized e_g_ electrons. As a result, the positive values of H_eff_^L1^, H_eff_^L2^ occur due to the paramagnetic excessive localized spins of Mn^4+^ ions in these layers (Figure 1b). For this reason, FMR for the L1, L2 lines at a fixed frequency are observed in weaker magnetic fields with the g-factor slightly exceeding 2.0. In this case, there is also a broadening of the FMR lines due scattering of spin excitations on holes carriers.

The symmetry of the L and R lines with respect to the L0 line (with the ratios of their intensities) is dictated by the requirement of the electro-neutrality of superlattice layers in the presence of differently charged manganese ions and charge carriers inside these layers. The change in such symmetry in the L-0-R neighboring layers leads to the electro-neutrality violation of superlattices, thereby forming the local electric polarization inside superlattices. In [15], the model of superlattices in the form of alternating L-0-R layers which explain the local electric polarization formation mechanism was proposed (see Figure 7 in [15]). That type of electric polarization was observed at low temperatures in GdMn_2_O_5_ along all crystal axes in electric field E ≠ 0 [16,17,18]. Similar polarizations were also observed in BiMn_2_O_5_ [19] and in R_(1−x)_Ce_x_Mn_2_O_5_ (X = 0, 0.2; R = Er, Tb) [21,22].

Let us now turn to the consideration of the influences of optical pumping power on ECMO superlattices, which are in the equilibrium state before optical pumping (Figure 2a). Pumping was carried out at T = 22 K by 15 ns pulses, with a peak power of ~0.5 MW, with a repetition rate of 10 Hz, for 1 min. Some preliminary studies of the weak optical pumping influences in ECMO were presented in [25].

Figure 2b shows against the background of equilibrium state of superlattice layers before optical pumping (black dots and curve (Figure 2a)) the set of FMRs from superlattices layers (red dots and curve) measured at the same temperature for a decreasing magnetic field at a rate of 1.2 kOe/min for 11 min, after switching off the optical pumping. The set of FMR lines is observed in the same resonance fields as before pumping, but with changed intensities of L0 and R1 lines. The intensities of other lines are nearly unvaried. In the equilibrium state (before pumping) the L0 line is almost twice as intense as the R1 line (Figure 2a), while the intensities these lines after pumping become equal (Figure 2b). We believe that excessive electrons (as compared with their initial concentration due to doping with Ce^4+^ ions) appear after the optical pumping in the L0 and R1 layers of superlattices due to relaxation of the electron-phonon band of manganese ions. The major part of these electrons localize on the Mn^4+^ ions (Mn^4+^ + e = Mn^3+^) in the L0 layers having equal numbers of Mn^3+^ and Mn^4+^ ions before pumping. In this case, excessive Jahn–Teller Mn^3+^ ions appear in the L0 layers. They deepen the wells inside these layers and bring their states closer to the R1 layer states. An intense exchange of electrons between the L0 and R1 layers (their tunneling) which makes electron concentrations in these layers equal arises (Figure 2b). The new state of superlattices, which is a long-lived one at low temperatures occurs, since it remains unchanged for at least 20 min after pumping is turned off.

Figure 2c shows the set of FMR lines after 16 h of natural sample heating up to 250 K and new cooling to 17.5 K (without new optical pumping). Black dots and lines correspond to an increasing magnetic field, red dots and lines are for a decreasing field. It can be seen that the L2 and L1 lines still do not change their intensities, both with increasing and decreasing magnetic field. At the same time, the intensities of the R2 and R1 lines sharply increase and the L0 line intensity somewhat decreases with increasing field.

It is natural to assume that the sample heating up to 250 K after pumping leads to a redistribution of excessive electrons generated by optical pumping between the L0, R1, R2 layers in such a way as to lower the superlattices energy. When the sample is heated, the kinetic energy of electrons should increase, which should cause their redistribution between layers not only due to tunneling between the L0 and R1 layers (Figure 2b) but also because of the hopping conductivity, which allows one to overcome higher barriers at the boundaries of other superlattice layers. In this case, the excessive electrons populate with the greatest probability deeper wells of the R2, R1, L0 layers (Figure 1b) and change their states so to ensure maximum intensities of FMRs of such layers (Figure 2c). This means that these layers should contain increased concentrations of ferromagnetic ion pairs with different valences, which are recharging by these excess electrons, thereby localizing them on these ions. The layers become dielectric and demonstrate intense and narrow FMR lines.

It should be noted that only a small part of the excessive electrons generated by optical pumping can transform Mn^4+^ ions into Mn^3+^ ions in the R2, R1 layers, which contain a significantly smaller amount of Mn^4+^ ions as compared with Mn^3+^ ions (Figure 1b). The main remaining part of such electrons transform Mn^3+^ ions into Mn^2+^ ions. As a result, R2 and R1 layers contain Mn^2+^ and Mn^3+^ ions, the Mn^2+^ ions being predominant. Recall that Mn^3+^ ions in the ground state consist of the triplet t_2g_ states, completely filled by three electrons, and of the degenerate doublet e_g_ filled by one electron. Mn^3+^ ions are Jahn–Teller ions and their amount is responsible for the contribution of Jahn–Teller interaction to the balance of internal interactions. The Mn^3+^ ion spins are equal to 2.0. The strongly magnetic Mn^2+^ ions have the completely filled t_2g_ and e_g_ states (their spins are equal to 5/2). A growth in the ferromagnetic moments of Mn^2+^ ions enhance the double exchange, but these ions are not Jahn–Teller ones. Recall that the electron accumulation in superlattices layers is caused by both Jahn–Teller interaction and the double exchange. The real states of superlattices in this case arise at the self-consisted changes of these two interactions. Figure 2c shows the state of such type for the case of magnetic field growth (the black points and line). In this case, the ferromagnetic moments of Mn^3+^-Mn^2+^ ion pairs increase, which should increase the double exchange between these pairs. As a result, the double exchange begins to prevail over Jahn–Teller interaction. In this case, the excess electrons are localized on the ferromagnetic Mn^3+^-Mn^2+^ ion pairs. When magnetic field decreases (Figure 2c, red points and lines) the role of double exchange in the balance of the main interactions decreases. The R2 line intensity becomes equal to that before pumping. The R1 line intensity, sharply decreasing in magnitude, still exceeds the L0 intensity. The next magnetic field increase (Figure 2d) resulted in the formation of the equilibrium state of the superlattices, but with lower intensities of their lines (compare Figure 2a,d).

Noteworthy is the fact that heating the sample to 250 K did not destroy the superlattices. There was only a change in their states. A new long-lived state with a new distribution of manganese ions of different valences in the layers of superlattices was formed, which significantly differed from their dynamically equilibrium state formed at low temperatures. This confirms the fact (discovered in an X-ray study at room temperature) that superlattices, changing their states, exist up to high temperatures. To restore the low-temperature dynamically equilibrium state of superlattices with a balance of basic interactions, it was necessary to cool the sample and carry out cycling in a magnetic field.

It is important that the process of formation of the equilibrium ground states of superlattices under the influence of successive changes in the external factors occurs through intermediate long-lived states. This is due to the fact that each step-by-step action causes a decrease in the superlattices energy with an intermediate balance of the main interactions and therefore is energetically favorable and long-lived.

It should be noted that there are less intense lines in lower magnetic fields near the main FMR lines in Figure 2. This can be explained by the fact noted above that in the crystal matrix of ECMO contains a smaller amount of Ce^3+^ ions along with impurity Ce^4+^ ions. Local regions of lattice distortions arise around the Ce^3+^ ions due to the presence of 6s^2^ lone electron pairs on their outer shells, which was observed by us in R_0.8_Ce_0.2_Mn_2_O_5_ (R = Er, Tb) [22]. Figure 2 shows a similar set of FMR lines for the regions near Ce^3+^ ions with lower intensities and in somewhat lower magnetic fields, i.e., similar superlattices are formed in these regions. This indicates that the lattice distortion inside the regions around the Ce^3+^ ions does not radically change the crystal matrix, maintaining the layered distribution of manganese ions of different valences along the c axis. In this case, the intensities of the superlattices FMR lines are lower than of the main lines due to the lower concentration of Ce^3+^ ions as compared with Ce^4+^. The magnetic fields in which these lines are observed also are slightly decreased due to differences of the internal interactions that form their balance in the equilibrium state of such type superlattices in compare with the main superlattices (Figure 2a).

Let us now consider the situation for ECMO in the equilibrium ground state after more powerful optical pumping at low temperatures (Figure 5). Pumping was carried out at T = 17.5 K by 15 ns pulses, with a peak power of ~0.5 MW, with a frequency of 14 Hz for 2 min.

In contrast to relatively low-power pumping, the relaxation of optical excitation of electron–phonon band of manganese ions at more powerful pumping should produce a larger number of free electrons both in the matrix and in the superlattices layers. At any pumping level, these electrons prefer to being concentrated in the superlattices layers both in the regions, formed due to Ce^4+^ ions and in the local regions near the Ce^3+^ ions, which form wells of different depths in the initial crystal matrix. As was shown above, at less powerful pumping, all excessive free electrons, generated by pumping, were localized after cycling in a magnetic field on the ferromagnetic pairs of manganese ions with different valences in the superlattices layers on the zero-matrix background (Figure 2). The ferromagnetic orientation of the spins in the layers of the superlattices were provided by the double exchange. In contrast to a relatively low power pumping, the relaxation of optical excitations at more powerful pumping led to changes in both the superlattices states and matrix background (Figure 5).

In this case, excessive free electrons were supposed to be localized in the wells formed by the superlattices and change their states. However, as it turned out, at the first magnetic field increase, the excessive electrons were distributed between the superlattices layers in such a manner that the states of these layers and the absorption background of matrix abruptly changed (Figure 5a). As noted above, the states of dynamically equilibrated superlattice layers before optical pumping and at weak pumping (Figure 2d) were characterized by such a charge distribution in L and R layers of superlattices in relation to the L0 layers that the electrical neutrality of the superlattices was provided.

The change of the absorption background after more powerful optical pumping (Figure 5a) can be attributed to the fact that the change in properties of superlattices layers induced by pumping violates the electro-neutrality of the superlattices. It gives rise to the additional internal electric field (the local electric polarization inside superlattices) oriented perpendicularly to the superlattices layers along the c axis. As a result, at low temperatures, when there is ferroelectric ordering along the b axis in the crystal matrix, the additional polarization along the c axis arises in superlattices. The jumps in the electric polarization that must be screened by charge carriers occur at the superlattices boundaries with the ferroelectric crystal matrix. This causes a change in the absorption background of the matrix. Thus, only a part of the excess electrons under high-power pumping changes the superlattices layers states and violates their equilibrium states. Another part of such electrons is spent on screening of the polarization jumps at boundaries of the superlattices with matrix. As a result, the more powerful optical pumping self-consistently changes both the properties of superlattices layers and of crystal matrix near their boundaries.

As noted above, the FMR line intensities of superlattices individual layers are proportional to the number of ferromagnetic manganese ion pairs with different valences. This also characterizes the double exchange contribution to a balance of the basic interactions, which determine the superlattices layer states. Let us now trace the changes in superlattices layers properties upon magnetic field cycling after powerful pumping.

Figure 5a shows dependences of the microwave absorption intensities after a new high-power optical pumping at the first growth in the magnetic field from 0 to 20 kOe (H || a), for the equilibrium sample state (Figure 2d) before optical pumping. Temperatures of the sample in both these states (in the equilibrium state after weak pumping and in the initial state before high-power optical pumping) were low (17.5 K), and the crystal was not heated up in the interval between these two measurements. In Figure 5a a sharp change in the absorption background and a high noise are noteworthy. Against this background change, a complete set of FMR lines from superlattices layers is observed in the same magnetic fields as before pumping and at the weak pumping (Figure 2).

It is noteworthy that the background change of microwave absorption with increasing magnetic field (Figure 5a) occurs in the range of magnetic fields in which FMR lines from the superlattices layers are recorded. Indeed, in low fields (up to the L2 layer position), the matrix background increases slightly, as observed before and after weak pumping (Figure 2). Under high-power pumping near the L2 line, the background sharply decreases. As we move towards the R2 line, the background increases linearly, reaching the initial level near the R2 line (Figure 5a). This indicates that the absorption background change is due to changes in the states of the superlattices layers that appear during the first magnetic field growth immediately after high-power pumping (Figure 5a).

Already with the first magnetic field, decrease (Figure 5b) the absorption background under FMR lines of the superlattices layers becomes zero, and only in the initial fields there is a certain residual background drop. In that case, a redistribution of the FMR line intensities occurs as shown in Figure 5b. Quite a sharp background drop in the lowest fields and its gradual decrease up to the maximal field also occur during the second growth in magnetic field (Figure 5c). As in the case of weak pumping, it becomes possible to form the dynamically equilibrium state of superlattices with a zero undistorted absorption background (Figure 5d) due to magnetic field cycling (at the third rise of magnetic field).

It is important to note that the properties of dynamically equilibrium superlattices formed by cycling in a magnetic field turned out to be similar in all cases: before pumping (Figure 2a), after weak pumping (Figure 2d), and after more powerful pumping (Figure 5d). In all this cases, all FMR lines of superlattices layers were observed against the zero background of the matrix, and the intensities of the FMR lines of the LO layers exceeded the intensities of both the left (L1, L2) and right (R1, R2) lines. A sharp violation of the intensity ratios of the FMR lines in comparison with the dynamically equilibrium state was observed at weak pumping after heating of the sample to 250 K, but the zero background of the matrix was not violated. It was only after the high-power pumping that there were sharp changes in both the FMR lines intensities and the matrix background. We attribute the sharp change in the matrix background to the violation of the electro-neutrality of the superlattices and to the appearance in the superlattices of electric polarization, oriented along the c axis of the crystal. In this case, cycling in a magnetic field also restored the dynamically equilibrium state of the superlattices against a zero background. Thus, the main criterion for breaking the electro-neutrality of superlattices is the distortion of the background of the crystal matrix.

As you can see from Figure 5d, along with narrow intense FMR lines, the additional FMR lines with only slight lower intensities are observed near the main FMR lines in the magnetic fields somewhat lower than for the main resonance lines. We attribute these additional lines to the superlattices formed in the regions near Ce^3+^ ions, which are also more actively populated by electrons under the high-power optical pumping.

Figure 6 shows temperature dependences of the line intensities of superlattices layers for the dynamically equilibrium state formed by the magnetic field cycling after high-power optical pumping. As temperature increases, the stable energetically favorable state of superlattices after high-power pumping is gradually destroyed up to 115 K. At the same time, such a state in this crystal before pumping disappeared near 85 K. Note that we are talking about the temperatures of existence of low-temperature dynamically equilibrium states of superlattices. We recall that higher-temperature states of superlattices with disturbed equilibrium states of superlattices can manifest themselves at higher temperatures. Striking examples of such manifestations are the existence of anomalously intense lines R1, R2 up to a temperature of 250 K (Figure 2c), as well as intense two right-hand lines of superlattices layers in an X-ray structural study (Figure 1c) at room temperature.

## 4. Conclusions

Thus, we have found that optically induced control of properties of superlattices in Eu_0.8_Ce_0.2_Mn_2_O_5_ multiferroic containing Mn^3+^ and Mn^4+^ ions is possible. These superlattices are formed in the initial as-grown crystals due to self-organization caused by a finite probability of electron tunneling between pairs of Mn^3+^ and Mn^4+^ ions located in neighboring crystal layers perpendicular to the c crystal axis. The superlattices are the semiconductor heterostructures with multiferroic properties and occupy a small crystal volume. The layers of superlattices contain ferromagnetic pairs of manganese ions Mn^3+^ and Mn^4+^ in various ratios. A set of ferromagnetic resonances was recorded from individual ferromagnetic layers of the superlattices, the parameters of which allowed us to define properties of the superlattice as a whole. The cycling of as-grown crystals in the magnetic field allows forming the dynamic equilibrium states of heterostructures–superlattices under a balance of strong competing interactions (double exchange, Jahn–Teller and Coulomb interactions). The superlattices layers properties in such states correlate with each other in such a way that it achieves the electrical neutrality of the entire superlattices. It was found that the properties of heterostructures–superlattices can be controlled by varying the temperature and magnetic field, and by optical pumping of various powers. The changes in the superlattice states were achieved by changing the concentration of manganese ion pairs with different valences in individual layers of superlattices. The most effective method of control of superlattice layer states was optical pumping. It was found that, depending on the optical pumping power, it was possible to control not only the ferromagnetic properties of superlattices–heterostructures but also their electric polar properties. Subsequent magnetic field cycling allowed us to restore the superlattice state to a state similar to the equilibrium one before pumping. After sufficiently powerful optical pumping, these restored states existed up to higher temperatures than equilibrium states in as-grown multiferroics. As a result, unlike artificially prepared, homogeneous bulk nanomaterials, the new nanomaterials on the example of the multiferroics–manganites Eu_0.8_Ce_0.2_Mn_2_O_5_ was studied. In these materials, the ferromagnetic regions in the form of superlattices (semiconductor heterostuctures) are formed in the antiferromagnetic matrix due to self-organization in the strong competing internal interactions. These superlattices, existing at wide temperature interval from the low temperatures up to room temperature, occupy the small volume in crystal matrix, but they determine magnetic and ferroelectric properties of the object as a whole at temperatures above the multiferroic ordering temperatures.

Some words about possible practical applications. The multiferroic under study belongs to the so-called type II multiferroics, in which ferroelectric ordering at temperatures of 30–35 K is induced by magnetic ordering with close temperatures of 40–45 K. The proximity of the ordering temperatures provides an anomalously strong magnetoelectric coupling. As a result, an attractive idea arises regarding the possibility of controlling the magnetic properties by an electric field, and vice versa, the electric properties by a magnetic field. A significant disadvantage is low temperatures. However, the presence of the charge ordering Mn^3+^ and Mn^4+^ ions and the finite probability of e_g_ electrons tunneling between these ions pairs lead to spontaneous (due to self-organization) formation in them of the local regions with properties similar to the initial matrix crystal, but existing at significantly higher temperatures, up to room temperature. The X-ray spectra, presented above, confirm the existence of such local regions at room temperature. In these regions, frozen superparaelectric and superparamagnetic states with strong magnetoelectric coupling arise for which the possibility of mutual control of their properties by the magnetic and electric fields is possible up to high temperatures (sometimes higher than room temperature). This was experimentally demonstrated in RMn_2_O_5_ and in doped R_0.8_Ce_0.2_Mn_2_O_5_ (R = Eu, Gd, Bi, Er, Tb) also [16,17,18,19,20,21,22]. Another factor that is important for practical applications is that the superlattices are formed due to self-organization. In that case, there are no inhomogeneities at their boundaries associated with the artificial fabrication of nanomaterials. We believe that multiferroics—manganites with local regions in the form of semiconductor-heterostructures and superlattices—can find possible application in spintronics, in the development of memory cells. We did not develop specific technical proposals.

## Figures and Tables

**Figure 1 nanomaterials-11-01664-f001:**
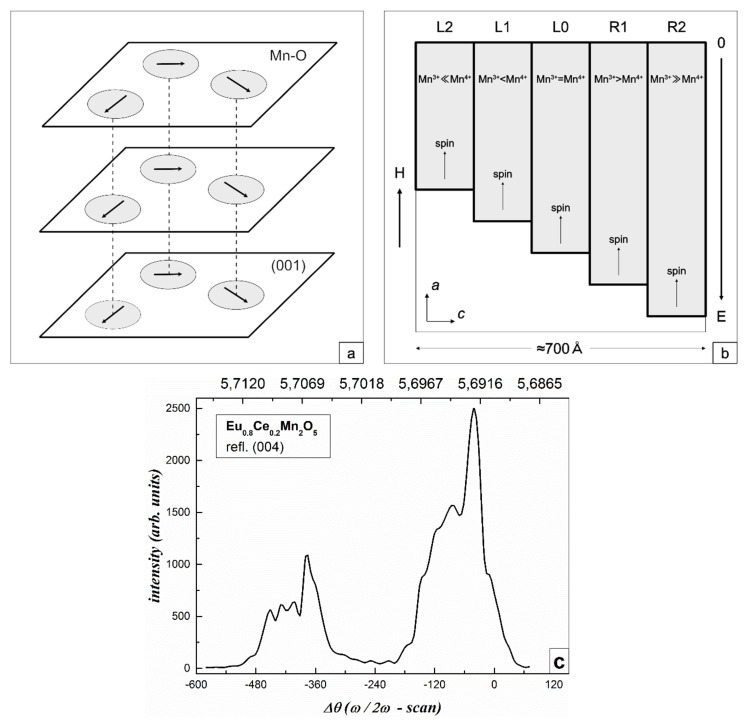
(**a**) Schematic representation of ferromagnetic 1D superlattices (filled areas) located in the initial crystal matrix (white field), magnetic field is H = 0. (**b**) Schematic representation of one of such superlattices consisting of L_N_ ferromagnetic layers perpendicular to the c axis with different concentrations of Mn^3+^-Mn^4+^ ion pairs and e_g_ electrons located in wells of different depths (filled regions), with different energies E. The dimensions of the superlattice regions in ECMO are ≈700 Å [10,13]. (**c**) Angular distribution of Bragg 004 *CuKα*1 reflection intensity for ECMO. The upper axis is lattice constant d along c axis in Å.

**Figure 2 nanomaterials-11-01664-f002:**
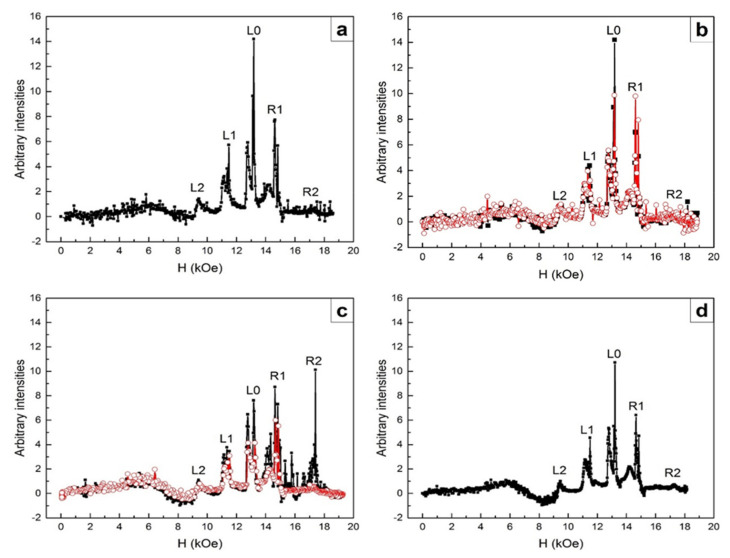
Intensity distribution of the FMR ECMO lines at frequency 34.5 GHz. Amplifier sensitivity 5 µV. H || a. The magnetic field variation rate is 1.2 kOe/min. (**a**) for as-grown sample, after the third cycle of magnetic field increase (dynamically equilibrated state of a 1D superlattice). T = 20 K. (**b**) The dynamically equilibrated state of the superlattice (black curve, Figure 2a) is compared with the intensity of distribution of FMR lines after optical pumping by 15 nsec pulses, with a power of 0.5 MW and repetition rate of 10 Hz, during 1 min. Decrease in magnetic field H (red curve). T = 22 K. (**c**) Change in the state of the sample after 16 h of its natural heating up to 250 K and its subsequent cooling to 17.5 K, at an increase in H (black curve) and a subsequent decrease in H (red curve). (**d**) Recovery of a state similar to the initial dynamic equilibrium state (Figure 2a) in the third cycle of magnetic field increase. T = 17.5 K.

**Figure 3 nanomaterials-11-01664-f003:**
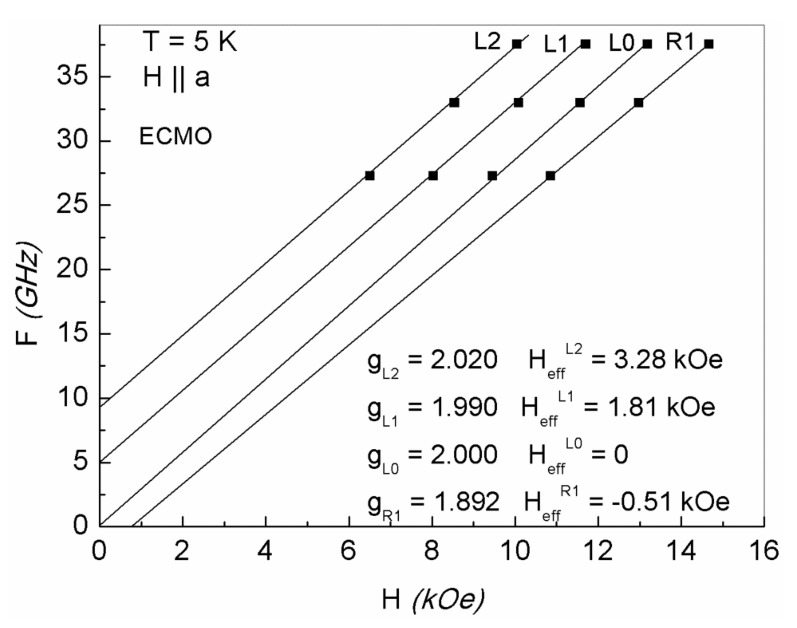
FMR frequencies versus magnetic field. FMR parameters of superlattice layers described by Equation (1) for ECMO at 5 K.

**Figure 4 nanomaterials-11-01664-f004:**
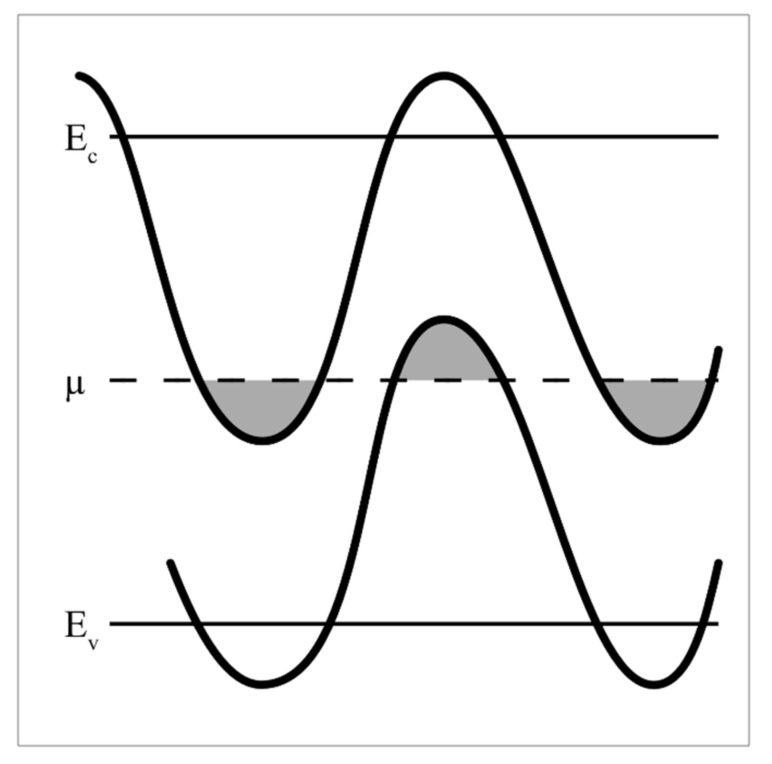
Schematic energy diagram of the L0 layers in the fully compensated semiconductor state. The upper and lower straight lines (E_c_ and E_v_) indicate the undistorted positions of the conduction band bottom and valence band top, respectively. The middle line µ corresponds to the Fermi level. The meandering lines are edges of the E_c_ and E_v_ bands modulated by the deformed potentials produced by charged impurities (Mn ions with different valences). The drops occupied by carriers (electrons and holes) are shaded.

**Figure 5 nanomaterials-11-01664-f005:**
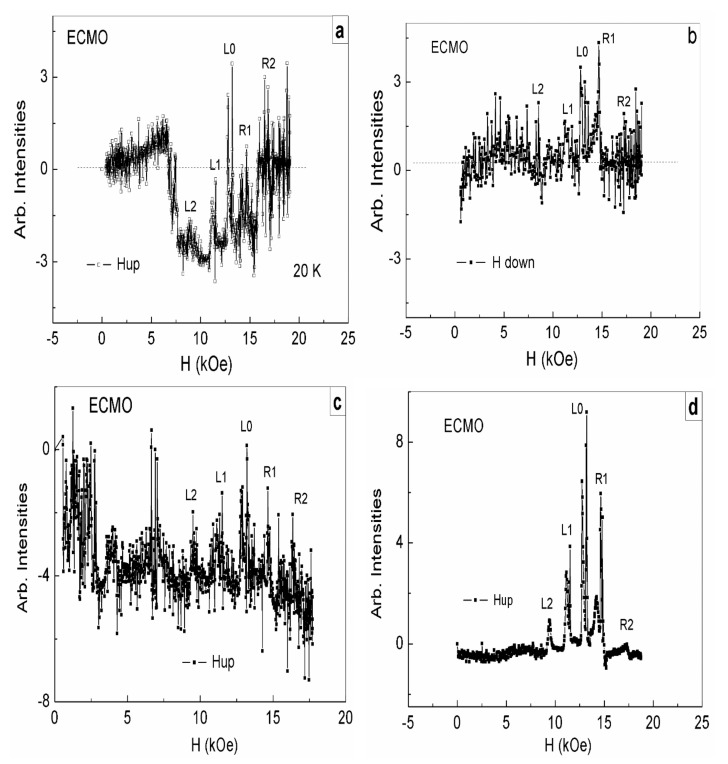
Distribution of intensities of observed FMR lines for the same ECMO sample at frequency 34.5 GHz. Amplifier sensitivity 5 µV. H || a. The rate of magnetic field change is 1.2 kOe/min., T = 17.5 K. (**a**) Changing in the sample state, which was at equilibrium state (Figure 2d), after powerful pumping, at the first magnetic field growth. (**b**) State of the superlattice layers after a subsequent decrease in the magnetic field. (**c**) State of the superlattice layers after the second increase in the magnetic field. (**d**) Recovery of the state similar to the initial dynamic equilibrium state of the sample (Figure 5d) in the third cycle of magnetic field increase.

**Figure 6 nanomaterials-11-01664-f006:**
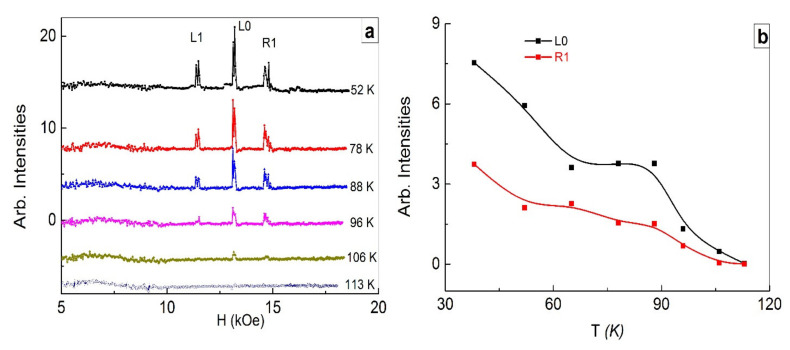
Intensities of the entire set of FMR lines from layers of superlattices for a dynamically equilibrium state of superlattices after high-power optical pumping (**a**). Temperature dependences of L0 and R1 line intensities (**b**).

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
