# Peer review of "Optical Control of Superlattices States Formed Due to Electronic Phase Separation in Multiferroic Eu0.8Ce0.2Mn2O5"

_nanomaterials, 2021, doi:10.3390/nano11071664_

Round 1

Reviewer 1 Report

In this paper, the authors report the effect of optical pumping and magnetic field on properties of phase separation nanodomains for Eu0.8Ce0.2Mn2O5, which at low temperatures are multiferroic semiconductor heterostructures in the form of 1D superlattices. This work contains interesting results and warrant for publication after the following improvements:

The abstract should be improved to indicate that ferromagnetic resonances (FMR) were used to investigate the effect of optical pumping at various powers on properties of ECMO superlattice layers in the dynamically equilibrated states.

The characterizations of the single crystals Eu0.8Ce0.2Mn2O5 should be represented in this paper, such as the results of X-ray phase analysis and X-ray fluorescence technique.

The unit for magnetic field should be the same, 2 T on page 4 should be changed to 20 kOe.

Fig. 2b, only half of H(kOe) appears.

Author Response

Response to Reviewer 1 Comments

Many thanks to Reviewer 1 for reading the manuscript and comments.

In this paper, the authors report the effect of optical pumping and magnetic field on properties of phase separation nanodomains for Eu0.8Ce0.2Mn2O5, which at low temperatures are multiferroic semiconductor heterostructures in the form of 1D superlattices. This work contains interesting results and warrant for publication after the following improvements:

Point 1: The abstract should be improved to indicate that ferromagnetic resonances (FMR) were used to investigate the effect of optical pumping at various powers on properties of ECMO superlattice layers in the dynamically equilibrated states.

Response 1: We include in Abstract the next insert “Sets of ferromagnetic resonances were observed from individual layers of superlattices. Their features give rise information on properties of these layers and of a superlattice as a whole. The differences in the parameters of these resonances were due to different distributions of Mn3+ and Mn4+ ions in individual superlattices layers. It has been found that optical pumping having different powers allows us to control of multiferroic properties of superlattices layers by changing their magnetic and electric properties.”

Point 2: The characterizations of the single crystals Eu0.8Ce0.2Mn2O5 should be represented in this paper, such as the results of X-ray phase analysis and X-ray fluorescence technique.

Response 2: We have included in Section II. Objects and research methods the following text. “The crystal symmetry and composition were determined by the X-ray phase analysis and X-ray fluorescence technique, respectively. X-ray diffraction studies were performed for several samples of ECMO at room temperature. The angular intensity distribution of 004_CuKα1 Bragg reflections was detected in the three-crystal regime with the θ−2θ scan having a resolution of 10''. This study of ECMO crystals have shown that ECMO has a single-crystal structure with layered superstructures. It was define the period of ECMO layered superstructure ~ 700 Å. X-ray fluorescence technique, made it possible to determine the concentration of Ce ions in ECMO (Eu0.8Ce0.2Mn2O5).”

Point 3: The unit for magnetic field should be the same, 2 T on page 4 should be changed to 20 kOe.

Response 3: That proof was included. 

Point 4:  Fig. 2b, only half of H(kOe) appears.

Response 4: That proof was included.

Reviewer 2 Report

The effect of optical pumping and magnetic field on properties of phase

 separation nanodomains for Eu0.8Ce0.2Mn2O5 have been studied in this manuscript. It is interesting and valuable to the readers in the field. But the authors should revise it by considering the following comments and suggestions.

(1) In Line 413, “We believe that the dynamically equilibrated state formed during the magnetic field cycling no longer produces an additional perpendicular electric polarization inside the superlattice layers and does not violate the charge neutrality”, it is suggested that the authors clarify more about the underline physical mechanism.

(2)The English writing needs to be improved, such as the sentence “It has been found that optical pumping having different powers allows control of multiferroic properties of superlattice layers by changing their magnetic and electric properties”

Author Response

Response to Reviewer 2 Comments.

Many thanks to Reviewer 2 for reading the manuscript and Comments.

Comments and Suggestions for Authors

The effect of optical pumping and magnetic field on properties of phase separation nanodomains for Eu0.8Ce0.2Mn2O5 have been studied in this manuscript. It is interesting and valuable to the readers in the field. But the authors should revise it by considering the following comments and suggestions.

Point 1: In Line 413, “We believe that the dynamically equilibrated state formed during the magnetic field cycling no longer produces an additional perpendicular electric polarization inside the superlattices layers and does not violate the charge neutrality”, it is suggested that the authors clarify more about the underline physical mechanism.

Response 1: It is important to note, that the properties of dynamically equilibrium superlattices formed by cycling in a magnetic field turned out to be similar in all cases: before pumping (Fig.2a), after weak pumping (Fig.2d) and more powerful pumping (Fig.5d). In all this cases, all FMR lines of superlattices layers were observed against the zero background of the matrix, and the intensities of the FMR lines of the LO layers exceeded the intensities of both the left (L1, L2) and right (R1, R2) lines. A sharp violation of the intensities ratios of the FMR lines in comparison with the dynamically equilibrium state was observed at weak pumping after heating of the sample to 250 K, but the zero background of the matrix was not violated. It was only after the high-power pumping there were sharp changes in both the FMR lines intensities and the matrix background. We attribute the sharp change in the matrix background to the violation of the electro-neutrality of the superlattices and to the appearance in the superlattices of electric polarization, oriented along the c axis of the crystal. In this case, cycling in a magnetic field also restored the dynamically equilibrium state of the superlattices against a zero background. Thus, the main criterion for breaking the electro-neutrality of superlattices is the distortion of the background of the crystal matrix (Last paragraph on Page 15 – with continued at the beginning of page 16).

Point 2: The English writing needs to be improved, such as the sentence “It has been found that optical pumping having different powers allows control of multiferroic properties of superlattice layers by changing their magnetic and electric properties”

Response 2: That text now is “It was found that, depending on the optical pumping power, it was possible to control not only the ferromagnetic properties of superlattices — heterostructures, but also their electric polar properties.

Reviewer 3 Report

The paper is focused on the in-depth studies of the multiferroic compounds by the measurements of ferromagnetic resonances. Some parts of the discussion I found interesting. However, the paper has a severe drawback listed below, which I believe should be addressed before I can judge regarding the possibility of paper publishing in Nanomaterials.

  1. The English should be significantly revised, possibly, with the help of external English-speaking specialists. In current form, it is challenging to understand the content of the paper. Even the naming of the paper is difficult to understand. I suppose the phrase “phase separation nanodomains” should be exchanged to grammatically correct.
  2. The paper is written in specialized language, assuming that most readers are familiar with the problems of that particular material. Nevertheless, the discussed multiferroics are not break-through materials. As such, some words should be said about why they should be studied and how they can be implemented in applications.
  3. As we assume that the paper should be about nanomaterials, clarification should be given regarding why the discussed compound is nanomaterial or what properties can be designed at the nano-level. To be honest, there is some discussion about it in the manuscript, but this point should be highlighted very carefully.
  4. Though the discussion is constructed based on the idea of existing nanodomains, the nature of these nanodomains remains unclear. They are taken from the literature review as a concept and with no apparent support from the experimental findings. I think that the manuscript is better to be separated into the results and discussion parts. The specific focus of discussion should clarify the domain nature and their role in the material properties.

Finally, the paper contains many mistakes and misprints. The manuscript should be carefully read and revised.

Author Response

Many thanks to Reviewer 3 for reading the manuscript and Comments. This was undoubtedly help to us a better presentation of the results of our study.

Point 1. The English should be significantly revised, possibly, with the help of external English-speaking specialists. In current form, it is challenging to understand the content of the paper. Even the naming of the paper is difficult to understand. I suppose the phrase “phase separation nanodomains” should be exchanged to grammatically correct.

Response 1. Regarding the English language - we used the help of a colleague who knows the language better, who read our manuscript and made a number of comments that we took into account. We hope that as a result, the text of the manuscript has become clearer.

 We changed the title of the article to “Optical control of superlattices states formed due to

 electronic phase separation in multiferroics  Eu0.8Ce0.2Mn2O5.”

Point 2.  Though the discussion is constructed based on the idea of existing nanodomains,   the nature of these nanodomains remains unclear. They are taken from the literature review as a concept and with no apparent support from the experimental findings. I think that the manuscript is better to be separated into the results and discussion parts. The specific focus of discussion should clarify the domain nature and their role in the material properties.

Response 2. Primary experimental information on the properties of nanodomains in multiferroics - manganites, studied by us in the our previously published articles, presented as the initial basis for analyzing the situation in our new studies, which are the subject of the article. The results of a theoretical work (review) are also presented, in which a model of nanodomains is considered in LaAMnO3 (A = Sr, Ba, Ca), which are also realized in the studied by us multiferroics- manganites with charge ordering. All necessary references are given and the validity of applying this model to our situation demonstrated. We have inserted into the text of the article the next paragraph substantiating the general nature of the appearance of nano-regions in LaAMnO3 manganites and in studied by us multiferroics - manganites.  “Note that studied by us the multiferroics - manganites are similar in their properties to the manganites LaAMnO3 (A = Sr, Ca, Ba) detail investigated before [7-9], that also contain Mn3+ and Mn4+ ions in various ratios depending on the degree of doping. Both types of manganites belong to systems, which are characterized by the strong electron correlations, exhibiting instability of uniform magnetic and charge orderings with respect to the formation of magnetic and charge polarons in the phase-separated state. The phase separation regions with 100 Å - 1000 Å sizes formed. In these regions, arranged in the dielectric antiferromagnetic matrix, a ferromagnetic orientation of spins Mn3+ and Mn4+ ions arises due to double exchange. Such local regions, containing electrons, recharging Mn3+ and Mn4+ ions, tend to be located far from each other, decreasing the energy of the Coulomb repulsion. As a result, these regions occupy a small volume in the crystal matrix [7-9]. (Page 2, second paragraph from top).”

Point 3. The paper is written in specialized language, assuming that most readers are familiar with the problems of that particular material. Nevertheless, the discussed multiferroics are not break-through materials. As such, some words should be said about why they should be studied and how they can be implemented in applications.

Response 3. In the Introduction, we considered in detail the features of the multiferroics we are studying. This is a well-known class of multiferroics, in which ferroelectric ordering is induced by magnetic ordering and both orderings have close temperatures, which ensures the appearance of a large magnetoelectric bond. They contain the same number of different valences (Mn3+ and Mn4+) ions, i.e. possess by charge ordering. That are realized at low temperatures. Along with these multiferroics have also features, which are studied in this work and which are determine their properties in the more wide temperature interval. As already reported above, in the multiferroics- manganites, studied by us, an electronic phase separation occurs with the formation of nano-regions (superlattices). The nano-regions in RMn2O5 and R0.8Ce0.2Mn2O5 (as in manganites LaAMnO3 (A = Sr, Ca, Ba)) are formed due to the balance of the strong interactions (double exchange (with a characteristic energy of 0.3 eV), Jahn-Teller interaction (0.7 eV), and Coulomb repulsion (1 eV)) [5, 7-9]. For this reason, they exist in a wide range of temperatures, from low ones to above room temperature [10, 12-22], while having a large magneto-electric coupling. This is attractive for practical applications. We studied the nano-regions of electronic phase separation in RMn2O5 multiferroics and in doped R0.8Ce0.2Mn2O5 (R = Eu, Gd, Bi, Er, Tb) also [10-24]. Note that RMn2O5 and doped R0.8Ce0.2Mn2O5 crystals have the same Pbam central symmetry. Investigations of dielectric and magnetic properties, heat capacity, X-ray diffraction, Raman light scattering [10-15], electric polarization [15-22], and µSR studies [23, 24] were carried out.

Point 4. As we assume that the paper should be about nanomaterials, clarification should be given regarding why the discussed compound is nanomaterial or what properties can be designed at the nano-level. To be honest, there is some discussion about it in the manuscript, but this point should be highlighted very carefully.

Response 4.

As noted above,   sets of ferromagnetic resonances (FMR) were observed from individual layers of superlattices in EuMn2O5 (EMO) and Eu0.8Ce0.2Mn2O5 (ECMO). Their features give information on properties of these layers and of superlattices as a whole [12-14]. It was found that the differences in the parameters of individual FMR lines of superlattices were due to different distributions of Mn3+ and Mn4+ ions in the individual superlattices layers. The sizes of these superlattices were 900 Å (EMO) and 700 Å (ECMO).  As a result, the new object for study arises in which, unlike to artificially prepared, homogeneous bulk nanomaterials, ferromagnetic nano-regions in the form of superlattices are formed in the antiferromagnetic matrix due to self-organization. These superlattices occupied the small crystal volume, but their contributions are decisive for the properties of the crystal as a whole.

Point 5. Finally, the paper contains many mistakes and misprints. The manuscript should be carefully read and revised.

Response 5.

The manuscript have been carefully read and revised.

Round 2

Reviewer 3 Report

I acknowledge the authors' efforts on the manuscript correction. The language becomes better. However, I have more comments on the topic, and I ask the authors to perform further modifications.

  1. I recommend not to use confusing naming “phase separation nanodomains” and “phase separation nano-regions”.
  2. I couldn’t find X-ray diffraction spectra from the studied crystals. Please insert it in the paper.
  3. I do not think that reference is enough to approve a concept of the nano-regions existence in the new material with its own sintering pathway and crystal state. Please provide experimental approval of the nano-regions’ existence by some structural method.
  4. I think that phrase “This is attractive for practical applications.” is not enough to discuss applications in the introduction part. I don’t understand the practical importance of the studied material. Tc temperature is very low, as I understood. Please provide detailed insight about how these materials can be used in practice, i.e., the concept of devices, possible practical application of the discussed physical phenomena, etc. References should support all the statements.
  5. I don’t understand this new statement in the conclusions: “These superlattices occupy the small crystal volume, but their properties practically define the properties of the crystal as a whole”. What kind of properties of the crystal is controlled by the superlattices? This is indeed important, and I expect some validation of the functional properties change under external stimuli of the nano-regions.

Author Response

Comments and Suggestions for Authors
-Referee3-Round2.

  1. I recommend not to use confusing naming “phase separation nanodomains” and “phase separation nano-regions”.
  2. I couldn’t find X-ray diffraction spectra from the studied crystals. Please insert it in the paper.
  3. I do not think that reference is enough to approve a concept of the nano-regions existence in the new material with its own sintering pathway and crystal state. Please provide experimental approval of the nano-regions’ existence by some structural method.
  4. I think that phrase “This is attractive for practical applications.” is not enough to discuss applications in the introduction part. I don’t understand the practical importance of the studied material. Tc temperature is very low, as I understood. Please provide detailed insight about how these materials can be used in practice, i.e., the concept of devices, possible practical application of the discussed physical phenomena, etc. References should support all the statements.
  5. I don’t understand this new statement in the conclusions: “These superlattices occupy the small crystal volume, but their properties practically define the properties of the crystal as a whole”. What kind of properties of the crystal is controlled by the superlattices? This is indeed important, and I expect some validation of the functional properties change under external stimuli of the nano-regions.

     Our responses:

  1. Answer: Having described the origin of the electronic phase separation in ECMO in the manuscript text, we adopted the term "superlattices" for their further use in the article.
  2.  

  •  
  1. Common answers. 

      We have included in the article text about the results of X-ray study Fig. 1c (Fig. 1a in

      [10]) at the room temperature. They demonstrated two different regions – responses from

      the crystal matrix and superlattices. The superlattices region were in the form of the set of

      layered reflexes.

  1. Answer: [Text included in paper]. The multiferroic under study belongs to the so-called type II multiferroics, in which ferroelectric ordering at temperatures of 30-35 K is induced by magnetic ordering with close temperatures of 40-45 K. The proximity of the ordering temperatures provides an anomalously strong magnetoelectric effect. As a result, an attractive idea arises of the possibility of controlling the magnetic properties by an electric field, and vice versa, by electric properties by a magnetic field. A significant disadvantage is low temperatures. However, the presence of the charge ordering Mn3+ and Mn4+ ions and the finite probability of eg electrons tunneling between these ions pairs in ECMO lead to spontaneous (due to self-organization) formation in them of the phase separation local regions (superlattices) with properties different from the matrix properties. These phase separation regions exist until to significantly higher temperatures. The X-ray spectra, presented above, confirm the existence of such local regions at the room temperature. In these regions, frozen superparaelectric and superparamagnetic states with strong magnetoelectric coupling arise that give rise the possibility of mutual control of their properties by the magnetic and electric fields until high temperatures (sometimes higher the room temperatures).  This was experimentally demonstrated in in RMn2O5 and in doped R8Ce0.2Mn2O5 (R = Eu, Gd, Bi, Er, Tb) also [16-22]. Another factor that is important for practical applications is that the superlattices are formed due to self-organization. In that case, there are no inhomogeneity at their boundaries associated with the artificial fabrication of nanomaterials. We believe that multiferroics - manganites with local regions in the form of semiconductor - heterostructures (superlattices) can find possible application in spintronics, in the development of memory cells. In this paper, we did not set the task of developing special technical applications.
  2. The point is not that superlattices control the properties of the matrix. It really is not possible. The high-temperature ferroelectric and magnetic properties of the entire object are determined by the properties of superlattices, which are superparamagnetic and superparaelectric ones. Keyword: Determine the property of the entire object, not control the properties of the matrix.

          (Text in paper) As a result, unlike to artificially prepared, homogeneous bulk nanomaterials, the new nanomaterials on the example of the multiferroics - manganites Eu0.8Сe0.2Mn2O5 was studied. In this materials, the ferromagnetic regions in the form of superlattices (semiconductor heterostuctures) are formed in the antiferromagnetic matrix due to self-organization in the strong competing internal interactions. These superlattices, existing at wide temperature interval from the low temperatures up to room temperature, occupy the small volume in crystal matrix, but they determine magnetic and ferroelectric properties of the object as a whole at temperatures above the multiferroic ordering temperatures.  

Inclusions of the new text insertions are shown in red.

Round 3

Reviewer 3 Report

The authors addressed most of my comments and the paper can be published in the present form. 

Author Response

We express our deep gratitude to our reviewers.